

# On gauging finite subgroups

**Yuji Tachikawa**

Kavli Institute for the Physics and Mathematics of the Universe,
University of Tokyo, Kashiwa, Chiba 277-8583, Japan

## Abstract

We study in general spacetime dimension the symmetry of the theory obtained by gauging a non-anomalous finite normal Abelian subgroup $A$ of a $\Gamma$-symmetric theory. Depending on how anomalous $\Gamma$ is, we find that the symmetry of the gauged theory can be i) a direct product of $G = \Gamma/A$ and a higher-form symmetry $\hat{A}$ with a mixed anomaly, where $\hat{A}$ is the Pontryagin dual of $A$; ii) an extension of the ordinary symmetry group $G$ by the higher-form symmetry $\hat{A}$; iii) or even more esoteric types of symmetries which are no longer groups. We also discuss the relations to the effect called the $H^3(G, \hat{A})$ symmetry localization obstruction in the condensed-matter theory and to some of the constructions in the works of Kapustin-Thorngren and Wang-Wen-Witten.

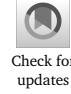

# 1   Introduction and summary

In this paper we start from a quantum field theory $T$ in general spacetime dimension $D$ with an ordinary symmetry group $\Gamma$ possibly with an anomaly. We pick a non-anomalous finite Abelian normal subgroup $A \subset \Gamma$. The aim of this note is to answer the following naive question: what is the symmetry group of the gauged theory $T/A$?

We know that the group $G = \Gamma/A$ is part of the story. We also know that the gauged theory has the $(D-2)$-form symmetry $\hat{A}_{[D-2]}$ based on the Pontryagin dual group $\hat{A}$ of $A$ [1, 2]. Here the notation $G_{[n]}$ stands for the group $G$ regarded as an $n$-form symmetry.

The question is how $G$ and $\hat{A}_{[D-2]}$ are mixed together. We will find that the result depends on the anomaly of $\Gamma$:

1. When $\Gamma$ is anomaly-free, the symmetry of $T/A$ is the direct product of $G$ and $\hat{A}_{[D-2]}$, with a mixed anomaly determined by the extension $0 \to A \to \Gamma \to G \to 0$.

2. When $\Gamma$ is anomalous but in a tame way as defined later in this paper, the symmetry of $T/A$ is an extension of $G$ by $\hat{A}_{[D-2]}$. In particular, $G$ is no longer a subgroup of the total symmetry. This extension is an example of a $(D-1)$-group, a group-like higher category.

3. When $\Gamma$ is anomalous in a more wild manner, the symmetry of $T/A$ is still composed of $G$ and $\hat{A}_{[D-2]}$, but the higher-category describing the total symmetry is no longer group-like.

Theories whose symmetry is given by higher categorical structures have also been discussed in the past e.g. [3–5]. Our point here is that such theories arise naturally if we perform even a rather innocuous operation of gauging a finite normal[1] Abelian group of a larger group.[2]

Our analysis can be thought of as a small extension of a few previous studies:

- The first case above generalizes and streamlines the technique of Gaiotto, Kapustin, Komargodski and Seiberg [9] where a combination of 0-form symmetry and 1-form symmetry with a mixed anomaly was turned into a non-Abelian 0-form symmetry. Starting from SU(2) gauge theory they found the dihedral group $(\mathbb{Z}_2 \times \mathbb{Z}_2) \rtimes \mathbb{Z}_2$ as the symmetry on a thermal circle. With our formula it is trivial to generalize it to SU($N$) gauge theory, for which we find $(\mathbb{Z}_N \times \mathbb{Z}_N) \rtimes \mathbb{Z}_2$.

---

[1]When we gauge a non-normal subgroup $A$, the story is more intricate. When $D = 2$, the resulting generalized symmetry is known [6] to be given by a fusion category, whose simple object is labeled by a pair $([g], \rho)$ where $[g] \in A \backslash \Gamma/A$ is a double coset and $\rho$ is an irreducible representation of $A \cap gAg^{-1}$, providing examples of non-group-like symmetry structures. It would be interesting to work out the higher-dimensional generalizations, we will not pursue this direction further in this note.

[2]The appearance of two-group symmetry when one gauges a continuous symmetry with a suitable mixed anomaly is also discussed in a recent paper [7]. See also a recent paper [8] for the discussions of finite gauge theories living on the boundary of gauged and ungauged Dijkgraaf-Witten theories.

- The first case above also generalizes the construction of Kapustin and Thorngren [10,11] to general spacetime dimensions and sheds a somewhat new light on a finite gauge theory construction in Witten [12] and Wang, Wen and Witten [13].

- The second case for $D = 3$ exhibits the effect called the $H^3(G, \hat{A})$ "symmetry localization anomaly" in the condensed-matter literature, see e.g. [14,15]. We find that this effect describes the extension of an ordinary group $G$ by a one-form symmetry $\hat{A}$.

- The second case also generalizes another work of Kapustin and Thorngren [16] from $D = 3$ to $D > 3$.

- Finally we note that for $D = 2$, the same question posed above was asked and answered in terms of the language of fusion categories long time ago, see e.g. [17–21]. For a recent review, see e.g. [22] by the author and Bhardwaj. In this case the analysis is completely general and allows non-Abelian or non-normal subgroup $A$.

In this note the discussions are phrased in terms of the anomaly of a $D$-dimensional theory on a spacetime $X_D$. That said, the general principle that the anomaly of a $D$-dimensional theory is captured by the corresponding $(D + 1)$-dimensional invertible topological phase [23–25] means that it is useful and essential to regard $X_D$ as a component of the boundary of a $(D+1)$-dimensional space $Y_{D+1}$.

**Organization of the note:**   The rest of the note is organized as follows. In Sec. , we start by analyzing what happens when we gauge an Abelian normal subgroup $A$ of an anomaly free symmetry $\Gamma$. We then study the case of the direct product $\Gamma = A \times G$ with a mixed anomaly. These two analyses turns out to be almost symmetric, and will be then be combined. We will also connect our analysis to the previous works of Kapustin and Thorngren [10, 11, 16], Witten [12] and Wang-Wen-Witten [13]. In Sec. 2.7, we discuss the generalization where both $G$ and $A$ are higher-form symmetries. We find that this generalization allows us to include 't Hooft defects in a natural manner. Finally in Sec. 3.3, we discuss subtler effects where the topological defects of the gauged theory contains objects which are no longer invertible. We have an appendix 4.3 where we review the required algebraic topology very briefly.

**Notations and conventions:**   We assume all groups to be finite for simplicity, although the non-gauged part $G$ can be readily made to be continuous. We write $G_{[n]}$ to mean that $G$ is an $n$-form symmetry, in the sense of [2]. We abbreviate $G_{[0]}$ as $G$. We use the corresponding lower case letter in bold, $\boldsymbol{g}$, for the flavor symmetry background for $G$. When $n = 0$, this is simply a $G$-bundle on the spacetime $M$, and when $n > 0$, this is an element of $H^{n+1}(M, G)$. Uniformly, one can say that the background $\boldsymbol{g}$ is a map from $M$ to the Eilenberg-Mac Lane space $K(G, n + 1)$; for more on this, see Appendix 4.3.

We reserve the letter $D$ to denote the spacetime dimension. We typically denote the spacetime as $X_D$ and consider it as a component of the boundary of a $(D + 1)$-dimensional space $Y_{D+1}$. The background fields such as $\boldsymbol{g}$ are also regarded as given on $Y_{D+1}$, not just on $X_D$.

We work in the convention that we only count Wilson lines as the topological operators we obtain for free in a $G$-gauge theory. Namely, 't Hooft defects require the coupling of the original theory to a singular $G$-gauge field, and in this paper we consider this to be an additional datum. In Sec. 2.7 we see that 't Hooft defects can be included in the framework as well.

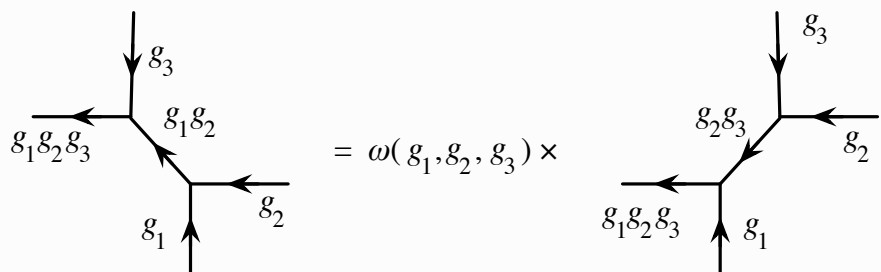

Figure 1: The anomalous phase is associated to the change in the topology of the domain walls on $X_D$ implementing the $G$ action.

## 2 Extension, anomaly and gauging

### 2.1 Preliminaries: anomaly, topological defects and group cohomology

Let us begin by recalling briefly how the anomaly of a finite group symmetry $G$ in spacetime dimension $D$ is given by an element $\omega \in H^{D+1}(G, U(1))$, and why it can be accounted for by coupling it to a $(D+1)$-dimensional bulk whose action is given by $\omega$. We illustrate it in the case $D = 2$.

We represent the background field for the $G$ symmetry in terms of topological domain walls. A domain wall is labeled by an element $g \in G$ such that crossing the wall implements the group action by $g$. Now, consider a situation where three domain walls $g_1$, $g_2$, $g_3$ merge to form a wall $g_1 g_2 g_3$. This can be done in two ways, see Fig. 1. Although they both describe the same background $G$ connection, a nontrivial phase $\omega(g_1, g_2, g_3) \in U(1)$ can be produced in an anomalous theory. In other words $\omega$ is a U(1)-valued 3-chain $\in C^3(G, U(1))$.

When we consider four domain walls $g_{1,2,3,4}$ joining to form a wall $g_1 g_2 g_3 g_4$, we can change how we join the walls either as

$$g_1(g_2(g_3 g_4)) \to g_1((g_2 g_3)g_4) \to (g_1(g_2 g_3))g_4 \to ((g_1 g_2)g_3)g_4 \tag{2.1}$$

or as

$$g_1(g_2(g_3 g_4)) \to (g_1 g_2)(g_3 g_4) \to ((g_1 g_2)g_3)g_4. \tag{2.2}$$

The consistency of these two procedures requires the condition

$$\partial \omega(g_1, g_2, g_3, g_4) :=$$
$$\omega(g_2, g_3, g_4) - \omega(g_1 g_2, g_3, g_4) + \omega(g_1, g_2 g_3, g_4) - \omega(g_1, g_2, g_3 g_4) + \omega(g_1, g_2, g_3) = 0, \tag{2.3}$$

where we use an additive notation for $U(1) = \mathbb{R}/\mathbb{Z}$. This means that $\omega$ is a U(1)-valued 3-cocycle $\omega \in Z^3(G, U(1))$. In general dimensions, $\omega$ is a U(1)-valued $(D+1)$-cocycle $\omega \in Z^{D+1}(G, U(1))$.

To see that this anomaly can be accounted for by coupling it to a bulk $(D+1)$-dimensional theory, let us use the dual triangulation instead of the domain walls to visualize the setup. Then two domain wall junctions on $X_D$ shown in Fig. 1 and the change between them can be depicted as in Fig. 2, i.e. due to an attachment of a $(D+1)$-simplex. Namely, we regard the domain walls and the junctions not to be defined only on $X_D$ but also to extend into the $(D+1)$-dimensional bulk $Y_{D+1}$. The introduction of an additional simplex to $Y_{D+1}$, which does not change the $G$-bundle, does change how the domain walls in the bulk join on the boundary. Then the phase $\omega(g_1, g_2, \ldots, g_{D+1})$ can be thought of as the Boltzmann weight associated

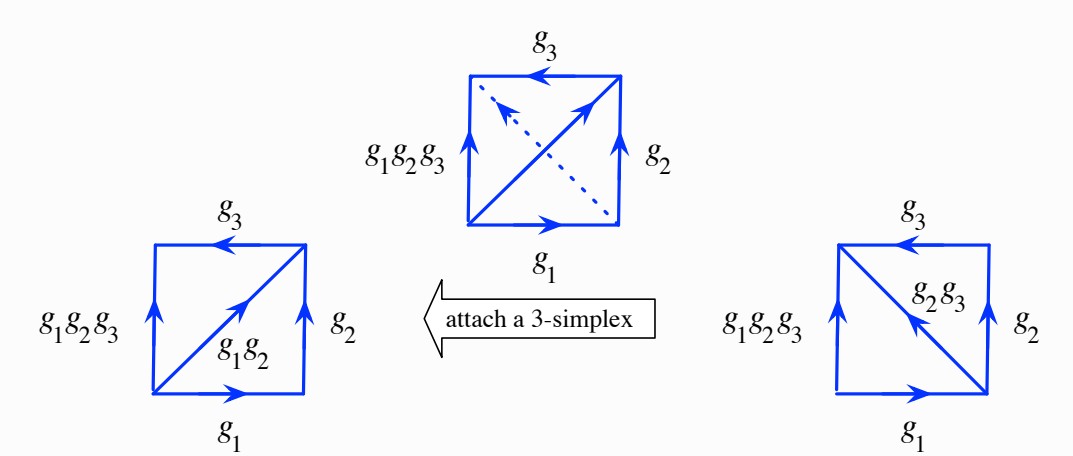

Figure 2: The anomalous phase can be thought of as due to attaching a $(D+1)$-simplex of the bulk theory on $Y_{D+1}$.

to this simplex of the $(D+1)$-dimensional symmetry-protected-topological (SPT) phase with $G$ symmetry on $Y_{D+1}$. From this viewpoint, the constraint (2.3) and its generalizations to arbitrary dimensions $D+1$ guarantee that the partition function assigned to a manifold with a triangulation does not depend on the choice of the triangulation; see e.g. [26,27] for a careful combinatorial treatment for low dimensional cases. The bulk theory is essentially the theory introduced by Dijkgraaf-Witten [28] but $G$ is now considered as a background field here, not as a dynamical field to be summed over.

We also note the following: suppose we are given a $D$-chain $L \in C^D(G, U(1))$. Then we can add to the action a term[3]

$$\int_{X_D} L(\boldsymbol{g}) \in U(1), \tag{2.4}$$

where $\boldsymbol{g}$ is the background $G$-bundle represented by the domain walls. This operation corresponds to the addition of contact and/or counter terms in the case of continuous flavor symmetry.

For example, for the figure on the left hand side of Fig. 2, the total additional coupling is $L(g_1, g_2) + L(g_1 g_2, g_3)$. Similarly, on the right hand side, the total additional coupling is $L(g_1, g_2 g_3) + L(g_2, g_3)$. This means that the anomaly $(D+1)$-chain $\omega$ is shifted to

$$\omega \to \omega' = \omega + \partial L, \tag{2.5}$$

where

$$\partial L(g_1, g_2, g_3) := L(g_2, g_3) - L(g_1 g_2, g_3) + L(g_1, g_2 g_3) - L(g_1, g_2). \tag{2.6}$$

This means that only $\omega \in H^{D+1}(G, U(1))$ matters up to the addition of the counter terms. That said, in an actual computation using this approach we are required to pick a particular cocycle representing $\omega$. For example, in Sec. 2.6, we will need to treat a trivial theory with $G$-symmetry with the anomaly cocycle $\omega = 0$ as a theory with the anomaly cocycle $\omega' = \partial L$ for a nontrivial $L$, by adding this contact term $L$.

Now, suppose we are given a $D$-dimensional $G$-symmetric theory $T$ with the anomaly $\omega$, and we want to gauge a subgroup $H \subset G$. What is the condition imposed? We represent the

---

[3]Or to be more precise, we should have said "we can multiply the exponentiated action by the term (2.4)" since this is a U(1)-valued phase.

$H$-bundle again using domain walls and their junctions. At each junction where two walls $h_1$ and $h_2$ meet and form the wall $h_1 h_2$, we assign the Boltzmann weight $\mu(h_1, h_2)$. In order for the total action to be consistent under the change in the topology of the domain wall, we need the condition

$$\partial \mu = \omega|_H. \tag{2.7}$$

This means that the element $\omega \in H^{D+1}(G, U(1))$ when pulled back to $H^{D+1}(H, U(1))$ should be trivial. In other words, $H$ should be anomaly free.

We note here that the difference of two possible solutions to (2.7) is an element of $H^D(H, U(1))$, but that there is no canonical solution in general, unless $\omega|_H$ itself happens to be zero. In this latter case, we can take $\mu = 0$. In the analyses we perform below, we almost always have this simplifying situation, so we take $\mu = 0$ unless otherwise mentioned.

## 2.2 From a nontrivial extension with a trivial anomaly

Suppose a theory $T$ has a 0-form finite group symmetry $\Gamma$ of the form

$$0 \to A \to \Gamma \to G = \Gamma/A \to 0. \tag{2.8}$$

We assume the whole group $\Gamma$ is anomaly free. Then we can gauge the subgroup $A$ with a trivial action. Denote the resulting gauged theory by $T/A$. What is the symmetry of $T/A$?

We assume $A$ is Abelian. Then the extension (2.8) is specified by an element

$$e \in H^2(G, A). \tag{2.9}$$

Our first result is the following:

> When the anomaly of $\Gamma$ is zero, $T/A$ has the symmetry $\hat{A}_{[D-2]} \times G$ whose total anomaly is given by
>
> $$\int_{Y_{D+1}} \hat{a} \cup e(\boldsymbol{g}) \in U(1), \tag{2.10}$$
>
> where $\hat{a} \in H^{D-1}(Y, \hat{A})$, $\boldsymbol{g}$ is the background $G$ connection, and $e(\boldsymbol{g}) \in H^2(Y, A)$ is the element defined by the extension class $e \in H^2(G, A)$ and $\boldsymbol{g}$.

Mathematically, $e(\boldsymbol{g})$ is determined as follows. $\boldsymbol{g}$ determines a map $Y \to BG$ which we denote by the same letter. $e \in H^2(G, A) = H^2(BG, A)$ can then be pulled back and gives an element $\boldsymbol{g}^*(e) \in H^2(Y, A)$. We denote the resulting element by $e(\boldsymbol{g})$ as a function of $\boldsymbol{g}$ determined by $e$.

When $D = 2$ this result was known in the mathematical literature [29, 30] for some time. When $D = 3$ this was derived essentially in [16] and also in [9] (see their Sec. 4.1 and Appendix B and C in particular) and an amazing dynamical application was given in the latter. Here we stick to pure formalities and give a derivation, essentially the one given in Appendix B of [9].

Let us first recall how $e$ determines the extension. We identify $\Gamma$ with $A \times G$ as a set. Then the product of $(0, g)$ and $(0, h)$ is given by $(e(g, h), gh)$. This $e(g, h)$ is the two-cocycle defining the extension. In terms of the domain wall operators implementing the action of $\Gamma$, this group law can be drawn as the figure on the left of Fig. 3. We can represent these domain walls also as on the right of Fig. 3. Namely, we regard the walls to implement the action of $A \times G$, but now the codimension-2 junction locus of two domain walls $g, h \in G$ forming $gh \in G$ serves as a *boundary* for the domain wall labeled by $e(g, h) \in A$.

This can be also phrased as follows. When $e = 0$, the background field $\boldsymbol{a}$ for the symmetry $A$ is an element in $H^1(X, A)$. When $e \neq 0$, $\boldsymbol{a}$ is not quite a cocycle, since the domain walls can now have boundaries. Instead, $\boldsymbol{a}$ is a cochain in $C^1(X, A)$ such that

$$\partial \boldsymbol{a} = e(\boldsymbol{g}), \tag{2.11}$$

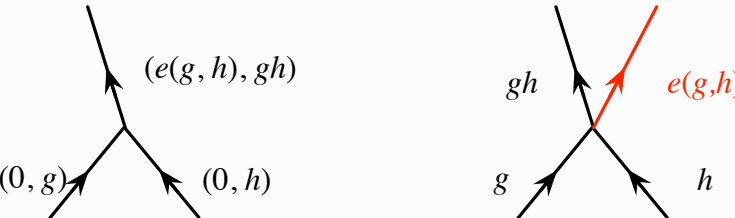

Figure 3: Domain walls for $\Gamma$ as domain walls for $A \times G$ such that domain walls of $A$ can have boundaries as determined by the extension class $e$.

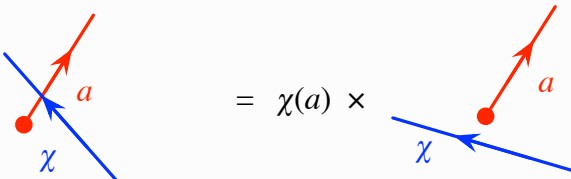

Figure 4: We have an additional phase when the Wilson line $\chi \in \hat{A}$ is moved across the boundary of the domain wall labeled by an element of $A$.

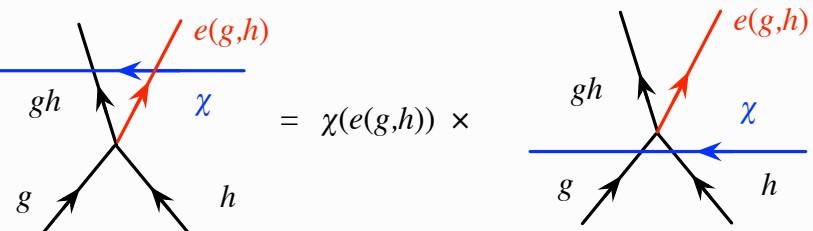

Figure 5: Combining the two facts, we find that moving the $\chi$ wall across the junction of $g$ and $h$ forming $gh$ produces a phase.

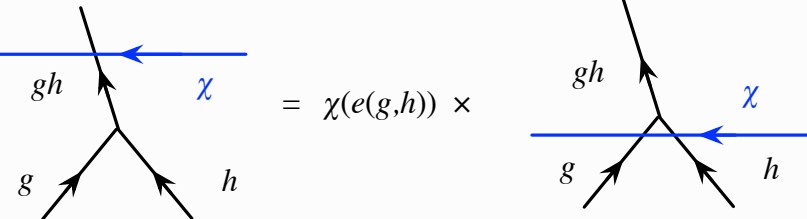

Figure 6: Summing over the $A$-walls, we now have a mixed anomaly between $\hat{A}$ and $G$.

where $\boldsymbol{g}$ is the background $G$ bundle on $X$, and $e$ is now thought of as a degree-2 characteristic class valued in $A$. Note that the latter is an analogue of the standard Green-Schwarz coupling: $dH = \operatorname{tr} F \wedge F$. In the absence of the Green-Schwarz effect, $H$ is a closed 3-form. However, in the presence of the Green-Schwarz coupling, $H$ is not quite closed, but its failure is given by a characteristic class constructed from another field.

Now consider a Wilson line operator labeled by $\chi \in \hat{A}$. When we move a line labeled by $\chi$ across a codimension-2 locus where the domain wall $a \in A$ starts, we get an extra phase $\chi(a)$, see Fig. 4. Recalling that the junction of two walls labeled by $g$ and $h$ forming a wall labeled

by $gh$ is a boundary from which the domain wall $e(g,h) \in A$ emanates, we find that moving a wall labeled by $\chi$ across the junction point produces a phase $\chi(e(g,h))$, see Fig. 5. Gauging $A$ means to sum over all possible domain walls implementing the symmetry $A$, and therefore $A$-walls are not considered backgrounds. We end up having the situation as shown in Fig. 6. Namely, we now have a mixed anomaly between the $(D-2)$-form symmetry $\hat{A}$ and the 0-form symmetry $G$, whose anomaly is given by (2.10).

### 2.3 From a trivial extension with a nontrivial anomaly

Next, let us consider the case when $\Gamma = A \times G$, or equivalently the extension class is zero, $e = 0 \in H^2(G,A)$. Let the total anomaly of $\Gamma = A \times G$ be given by

$$\int_{Y_{D+1}} \boldsymbol{a} \cup \tilde{e}(\boldsymbol{g}), \tag{2.12}$$

where $\boldsymbol{a} \in H^1(Y,A)$ and $\tilde{e} \in H^D(G,\hat{A})$. Our next claim is the following:

> *The symmetry of $T/A$ is an "extension of $G$ by $\hat{A}_{[D-2]}$"*
>
> $$0 \to \hat{A}_{[D-2]} \to \underline{\tilde{\Gamma}} \to G \to 0, \tag{2.13}$$
>
> *whose extension class is given by*
> $$\tilde{e} \in H^D(G,\hat{A}). \tag{2.14}$$

Here we put an underline to $\tilde{\Gamma}$ to emphasize that this is a mixture of an ordinary 0-form symmetry and a higher form symmetry. It should be possible to give a mathematical definition of this object $\underline{\tilde{\Gamma}}$ itself in terms of the concept of $(D-1)$-group[4], but for us it suffices to describe what are the background fields for this extension[5]. Namely, a background field for this symmetry on a spacetime $X$ is a pair $(\boldsymbol{a}, \boldsymbol{g})$, where

- $\boldsymbol{g}$ is a $G$ bundle on $X$, and

- $\boldsymbol{a}$ is a $(D-1)$-cochain valued in $\hat{A}$ such that $\partial \boldsymbol{a} = \tilde{e}(\boldsymbol{g})$.

With this understanding, the derivation is entirely analogous to the first case discussed above and is not repeated here. We also note again that the equation $\partial \boldsymbol{a} = \tilde{e}(\boldsymbol{g})$ can be thought of as a finite analogue of the Green-Schwarz effect.

### 2.4 Miscellaneous remarks

- In the language of the topological defects, the extension can be described as follows. Take a point $p$ in the spacetime $X_D$ where $D$ domain walls implementing elements $g_1, \ldots, g_D$ of $G$ meet and form the wall for $g_1 g_2 \cdots g_D$. Then, from this point, a defect line operator implementing an element $\hat{a}$ of $\hat{A}_{[D-2]}$ comes out, where $\hat{a}$ is given by $\hat{a} = \tilde{e}(g_1, \ldots, g_D)$.

---

[4] Here an $n$-group refers loosely to an $n$-category with a single object where every $k$-morphism is invertible in a suitable sense. As there are multiple subtly different notions of $n$-categories for $n \geq 2$, it is unwieldy to be specific. Note also that in group theory a $p$-group refers to a finite group whose order is a power of a prime $p$, but we are not particularly interested in those in this note.

[5] The situation is analogous to that of a non-commutative space $M$: We define the non-commutative algebra $\mathcal{A}$ first, and then say that $M$ is something whose algebra of functions is $\mathcal{A}$, defining $M$ indirectly.

- At a purely formal level, the 't Hooft anomaly itself can be thought of as an extension, as follows. Note that every QFT has a $U(1)_{[D-1]}$ symmetry, whose corresponding topological defect operator is simply a point operator given by an insertion of a complex number of absolute value 1. Then the 't Hooft anomaly $\in H^{D+1}(G, U(1))$ specifies an extension

$$0 \to U(1)_{[D-1]} \to \underline{\tilde{G}} \to G \to 0. \tag{2.15}$$

This consideration, however, does not buy us much, since the $U(1)_{[D-1]}$ symmetry acts in an entirely straightforward manner on the theory.[6]

- When $D = 3$, the resulting mixed symmetry is a 2-group, and the corresponding 4d TQFT was studied in [16]. An appearance of higher groups in physics was also noted in [3].

- The appearance of this extension in $D = 3$ in a theory with $\hat{A}$ 1-form symmetry and $G$ domain walls is sometimes called as the "$H^3(G, \hat{A})$ symmetry localization anomaly" in the cond-mat literature, see e.g. [14, 15]. But this is somewhat of a misnomer, since the element $\tilde{e}$ is a part of the group law in the theory $T/A$, and not the anomaly in the ordinary sense. We prefer to call it as the $H^3(G, \hat{A})$ phenomenon.

- In [15] it was pointed out that a system with a nonzero $\tilde{e} \in H^3(G, \hat{A})$ can be realized on a boundary of an $A$-gauge theory. This can be naturally understood from the point of view of this note: a system with nonzero $\tilde{e}$ phenomenon can be realized as an $A$-gauge theory of a theory with $\Gamma$ symmetry with an anomaly $\tilde{e}$. This latter theory can be realized on a boundary of a $\Gamma$-SPT. We gauge $A$ of the entire setup. Then the original theory with a nonzero $\tilde{e}$ phenomenon is realized on a boundary of an $A$-gauge theory.

- Another related point is as follows. When $D = 3$, there are many studies of TQFTs with a $G$ action. The mathematics is captured by a $G$-crossed modular tensor category $\mathcal{C} = \oplus_g \mathcal{C}_g$ where objects in $\mathcal{C}_g$ are the boundary lines of a domain wall implementing an action of $g \in G$. In this context people speak of the $H^3(G, \hat{A})$ anomaly/phenomenon/obstruction [31]: we start from an action of $G$ on a category $\mathcal{C}_{id}$ and asks if it is possible to extend this data to a genuine $G$-crossed category $\mathcal{C} = \oplus_g \mathcal{C}_g$. It is known that there is an obstruction which appears as the failure of the isomorphism of tensor products of three objects $x_{1,2,3} \in \mathcal{C}_{g_{1,2,3}}$. In a genuine category, we would like to have

$$x_1 \otimes (x_2 \otimes x_3) \simeq (x_1 \otimes x_2) \otimes x_3 \tag{2.16}$$

but in general we are forced to have

$$x_1 \otimes (x_2 \otimes x_3) \simeq ((x_1 \otimes x_2) \otimes x_3) \otimes \tilde{e}(x_1, x_2, x_3), \tag{2.17}$$

where $\tilde{e}(x_1, x_2, x_e)$ is an invertible object in $\mathcal{C}_{id}$. We define $\hat{A}$ to be the subcategory of the invertible objects in $\mathcal{C}_{id}$. Then $\tilde{e}$ defines a three-cocycle representing an element in $H^3(G, \hat{A})$, and this is an obstruction to the existence of the $G$-crossed tensor category $\mathcal{C} = \oplus_g \mathcal{C}_g$.

A higher-categorical framework for arbitrary topological defects in $D = 3$ was given in [4] and their gauging was studied in [5]. The point is that when this $H^3(G, \hat{A})$ obstruction is nonzero, the resulting topological defects are described by a higher-category more general than $G$-crossed modular tensor categories, and that such an example can be readily obtained by gauging a normal Abelian subgroup of a group with an anomaly.

It should also be noted that even when the $H^3$ obstruction vanishes, there is the next obstruction which takes value in $H^4(G, U(1))$ [31]. From the physics point of view, it

---

[6]The author thanks C. Córdova and N. Seiberg for discussions on this point.

is the anomaly of the $G$ symmetry of the 3d TQFT under consideration. These issues are treated in detail from the physics point of view in a paper by Benini, Córdova and Hsin [32].

## 2.5   From a nontrivial extension with a nontrivial anomaly

The discussions in Sec. 2.2 and Sec. 2.3 are symmetric; indeed they are the same when $D = 2$. We can in fact mix them, which is well known for $D = 2$ [29, 30, 33, 34].

Suppose a theory $T$ has the symmetry $\Gamma$ given by a group extension as above (2.8), with the extension specified by $e \in H^2(G, A)$. As in the discussions above, a background $\Gamma$ bundle $\gamma$ can be decomposed into a pair $(a, g)$ where $a$ is a 1-cochain valued in $A$ and $g$ is a background $G$ bundle so that $\partial a = e(g)$.

Given an element

$$\tilde{e} \in H^D(G, H^1(A, U(1))) = H^D(G, \hat{A}), \tag{2.18}$$

we try to use the same formula $a \cup \tilde{e}(g)$ (2.12) to define an element in $H^{D+1}(\Gamma, U(1))$. This is in general inconsistent, since

$$\partial(a \cup \tilde{e}(g)) = (e \cup \tilde{e})(g). \tag{2.19}$$

We need to require that

$$e \cup \tilde{e} = 0 \in H^{D+2}(G, U(1)). \tag{2.20}$$

Assuming this, we can pick an element $\omega$ in $C^{D+1}(G, U(1))$ such that

$$\partial \omega = e \cup \tilde{e} \in C^{D+2}(G, U(1)). \tag{2.21}$$

Then we can define an element of $H^{D+1}(\Gamma, U(1))$ by the formula

$$\alpha(\tilde{e}, \omega) := \int_{Y_{D+1}} (a \cup \tilde{e}(g) - \omega(g)). \tag{2.22}$$

Note that $\omega$ is defined up to additions of elements in $H^{D+1}(G, U(1))$.

Now, let us gauge $A$. The gauged theory $T/A$ has the symmetry $\tilde{\Gamma}_{[D-2,0]}$ (2.13) determined by $\tilde{e}$. The anomaly is given by

$$\tilde{\alpha}(e, \omega) := \int_{Y_{D+1}} (\hat{a} \cup e(g) - \omega(g)). \tag{2.23}$$

Summarizing, we have the following:

> Given $e \in H^2(G, A)$, $\tilde{e} \in H^D(G, \hat{A})$ satisfying $e \cup \tilde{e} = 0$ and an element $\omega \in C^{D+1}(G, U(1))$ such that $\partial \omega = e \cup \tilde{e}$, a theory with an ordinary symmetry
>
> $$0 \to A \to \Gamma \to G \to 0 \tag{2.24}$$
>
> specified by $e$ with an anomaly $\alpha(\tilde{e}, \omega)$ as in (2.22) and a theory with the mixed symmetry
>
> $$0 \to \hat{A}_{[D-2]} \to \tilde{\underline{\Gamma}} \to G \to 0 \tag{2.25}$$
>
> specified by $\tilde{e}$ with an anomaly $\tilde{\alpha}(e, \omega)$ as in (2.23) are exchanged by the gauging of $A$ and $\hat{A}$.

In terms of the Lyndon-Hochschild-Serre spectral sequence briefly reviewed in Appendix A.2, we can study the anomaly $H^{p+q}(\Gamma, U(1))$ in terms of $H^p(G, H^q(A, U(1)))$. In this language, the condition (2.20) is the vanishing $d_2(\tilde{e}) = 0$.

## 2.6   Relation to other constructions

To actually realize a theory $T/A$ with a nontrivial extension of a group by a higher-form symmetry such as (2.13), we need to have a theory $T$ with a finite group symmetry $\Gamma$ with a specified nontrivial anomaly. It is not obvious that such a theory $T$ actually exists, but luckily a construction for $D = 2, 3$ in terms of finite gauge theory was given by Kapustin and Thorngren [10, 11] which was further generalized to arbitrary dimensions in Sec. 3.3 of Witten [12] and Wang-Wen-Witten [13]. In fact their construction itself can be thought of as an example of the discussions so far. We would like to elaborate on this point.

### 2.6.1   Relation to Witten [12] and Wang-Wen-Witten [13]

We would like to reproduce the following statement from a paper by Witten [12] and another by Wang-Wen-Witten [13][7]:

> *Given a group $G$ and an anomaly $\omega \in H^{D+1}(G, U(1))$, suppose we are given an Abelian extension*
>
> $$0 \to A \to \Gamma \to G \to 0 \tag{2.26}$$
>
> *such that $\omega$ trivializes in $H^{D+1}(\Gamma, U(1))$. Denote by $T$ an almost trivial theory with $\Gamma$ symmetry as described below. Then the gauged theory $T/A$ has the symmetry $G$ with an anomaly $\omega$.*

In Sec. 5 of [13] an argument was given to find such an extension $\Gamma$ by a suitable finite Abelian $A$ for any $G$ and $\omega \in H^{D+1}(G, U(1))$ such that the pullback of $\omega$ to $\Gamma$ trivializes. We will give a mathematical proof in Sec. 2.7. For a while, we assume that such an extension is given.

We now specify the almost trivial theory $T$ to be used. We take a specific cocycle $\omega \in Z^{D+1}(G, U(1))$, and pull it back to $\Gamma$. We denote the resulting cocycle by the same symbol $\omega$. As we assumed that this trivializes in the cohomology, there is a $\mu \in C^D(\Gamma, U(1))$ such that $\omega = \partial \mu$. We now consider a completely trivial theory with $\Gamma$ symmetry, and use $\mu$ as the action, as explained in (2.4). This is the theory $T$ used in the statement above. Note that this theory $T$ is not completely trivial, in the sense that it has a nonzero anomaly cocycle $\omega \in Z^{D+1}(\Gamma, U(1))$, although $\omega$ is trivial in $H^{D+1}(\Gamma, U(1))$. However, the anomaly cocycle restricted to $A$ is trivial, since $\partial \mu|_A = \omega|_A = 0$. Therefore one can gauge $A$.

It is almost tautological from the construction that the gauged theory $T/A$ has the anomaly cocycle $\omega \in Z^{D+1}(\Gamma, U(1))$. In fact this is a special case of our discussion in Sec. 2.5. Indeed, the condition (2.21) is trivially satisfied since $\tilde{e} = 0$. Then we just restrict (2.23) to $G$ by setting $\hat{a} = 0$. We find that the anomaly is simply given by $\omega$ itself.

### 2.6.2   Relation to Kapustin-Thorngren [10, 11]

The argument above was mostly tautological, so let us re-analyze it in slightly more detail. This brings us closer to the point of view of the two papers by Kapustin and Thorngren [10, 11].

We use the Lyndon-Hochschild-Serre spectral sequence, briefly reviewed in Appendix A.2. As an assumption, we have a natural pullback $H^{D+1}(G, U(1)) \to H^{D+1}(\Gamma, U(1))$ under which $\omega$ is sent to zero. This pullback map is decomposed into steps in the LHS spectral sequence:

$$H^{D+1}(G, U(1)) = E_2^{D+1,0} \twoheadrightarrow E_3^{D+1,0} \twoheadrightarrow \cdots \twoheadrightarrow E_{D+2}^{D+1,0} = E_\infty^{D+1,0} \hookrightarrow H^{D+1}(\Gamma, U(1)), \tag{2.27}$$

---

[7]In their papers they used the symbols $(K, H, G)$ instead of our $(A, \Gamma, G)$. Also, their papers contain many more results, and here we are only scratching a tiny part of them.

where

$$E_{r+1}^{D+1,0} = E_r^{D+1,0} / \operatorname{Im} d_r. \tag{2.28}$$

Note that every arrow except the last is a surjection and the last is an inclusion.

**When $\omega$ is in the image of $d_2$:** Now, suppose that $\omega$ is sent to zero in the first step, i.e. $\omega$ is in the image of $d_2$. Since $d_2$ is a cup product with the extension class $e$, this means that there is an element $b \in E_2^{D-1,1} = H^{D-1}(G,\hat{A})$ such that $b \cup e = \omega$. With this assumption, $\omega$ should be a coboundary. Indeed, we see that $\partial(b(\boldsymbol{g}) \cup \boldsymbol{a}) = \omega(\boldsymbol{g})$, where $\boldsymbol{a}$ as always satisfies $\partial \boldsymbol{a} = e(\boldsymbol{e})$.

In the previous sections, we were always gauging $A$ of a theory $T$ with a trivial action but in a presence of an anomaly cochain $\omega$. When the anomaly cochain is zero and the action of the $A$ gauge theory is also zero, the setup is exactly as in Sec. 2.2, the symmetry is $\hat{A}_{[D-2]} \times G$ and the anomaly of the gauged theory is purely mixed and was given in (2.10). We do not have an anomaly purely of $G$ yet.

Now, let us "embed" the group $G$ into the mixed group $\hat{A}_{[D-2]} \times G$ by the formula

$$\boldsymbol{g} \to (b(\boldsymbol{g}),\boldsymbol{g}), \tag{2.29}$$

where the left hand side is a $G$ bundle, and the right hand side is a background of $\hat{A}_{[D-2]} \times G$, i.e. a pair of an element of $H^{D-1}(-,\hat{A})$ and a $G$-bundle. Note that when $D = 2$, an element $b \in H^1(G,\hat{A})$ actually gives a homomorphism $b : G \to \hat{A}$, and the operation (2.29) indeed comes from a homomorphism $G \to \hat{A} \times G$. Therefore an element $b \in H^{D-1}(G,\hat{A})$ is a natural higher version of this homomorphism. Plugging this in to (2.10), we see that the anomaly is now

$$\int_{Y_{D+1}} b(\boldsymbol{g}) \cup e(\boldsymbol{g}) = \int_{Y_{D+1}} \omega(\boldsymbol{g}). \tag{2.30}$$

This embedding (2.29), in the language of the $A$ gauge theory, can be realized as the coupling in the action of the form

$$\int_{X_D} b(\boldsymbol{g}) \cup \boldsymbol{a}. \tag{2.31}$$

Stated differently, if we gauge the original theory $T$ with the action (2.31), we get the gauged theory $T/A$ with an anomaly $\omega$ for the symmetry $G$.

**When $\omega$ is in the image of $d_3$:** Let us next consider the case when $\omega$ is killed at the second step, i.e. when it is in the image of $d_3$. For simplicity of the presentation, we restrict to the case $D = 2$. Then we have an element $\beta \in H^0(G,H^2(A,U(1))) = H^2(A,U(1))$ such that $d_2\beta = 0$ and $d_3\beta = \omega$.

For explicitness, we set $A = (\mathbb{Z}_n)^k$, and we can now match the data with the analysis of Sec. 3.1 of Kapustin and Thorngren [11]. In our notation, this goes as follows.

From the data $\beta \in H^2(A,U(1))$ we try the following expression as the action of the $(\mathbb{Z}_n)^k$ gauge field:

$$\int_{X_2} \sum_{i<j} \beta_{ij} \boldsymbol{a}^i \cup \boldsymbol{a}^j, \tag{2.32}$$

where $i, j = 1, \ldots, k$ labels the components of $(\mathbb{Z}_n)^k$ and $\beta_{ij}$ is now thought of as a $\mathbb{Z}_n$-valued skew-symmetric matrix. This is the standard form of the discrete torsion for $A = (\mathbb{Z}_n)^k$. With the extension $\partial \boldsymbol{a}^i = e^i(\boldsymbol{g})$, however, the coboundary of the integrand of (2.32) is

$$\partial(\sum_{i<j} \beta_{ij} \boldsymbol{a}^i \cup \boldsymbol{a}^j) = (\sum_{i,j} \beta_{ij} e^i \cup \boldsymbol{a}^j) + \sum_{i<j} \beta_{ij}(-\partial(\boldsymbol{a}^i \cup_1 e^j) + e^i \cup_1 e^j), \tag{2.33}$$

which is $\boldsymbol{a}$-dependent in general and therefore inconsistent.

To be able to rectify it, we need to require that the object $\beta_{ij}e^i$ is a coboundary, $\beta_{ij}e^i = \partial b_j$. Note that $\beta_{ij}e^i$ naturally is an element of $H^2(G, \hat{A}) = H^2(G, H^1(A, U(1)))$, and can be identified with $d_2\beta$ in the LHS spectral sequence. Therefore such $b_i$ exists if and only if $d_2\beta = 0$.

Assuming the existence of $b_i$, we then consider the action

$$\int_{X_2} (\sum_{i<j} \beta_{ij}(\boldsymbol{a}^i \cup \boldsymbol{a}^j + \boldsymbol{a}^i \cup_1 e^j(\boldsymbol{g})) + \sum_i b_i(\boldsymbol{g}) \cup \boldsymbol{a}^i). \tag{2.34}$$

The coboundary of the integrand is now

$$\omega = \sum_{i<j} \beta_{ij}e^i \cup_1 e^j + \sum_i b_i \cup e^i \in H^3(G, U(1)), \tag{2.35}$$

which can be identified with $d_3\beta$ and describe the anomaly of the symmetry $G$ of the gauge theory.

Kapustin and Thorngren also gave the analysis of the case $D = 3$, $A = \mathbb{Z}_n$ and described $d_4$ explicitly. It would be interesting to actually construct the LHS spectral sequence using their approach.

## 2.7 Existence of finite Abelian extension where the anomaly trivializes

Let us now prove[8] that for any $\omega \in H^{D+1}(G, U(1))$ for $D + 1 \geq 2$, there is a finite Abelian extension

$$0 \to A \to \Gamma \to G \to 0, \tag{2.36}$$

such that $\omega$ trivializes in $\Gamma$.

It is well-known [35, Sec. 6] that in the algebraic definition of group cohomology, we can restrict attention to normalized cochains, i.e. those cochains $c$ such that $c(g_1, \ldots, g_n) = 0$ if any of $g_i = 1$, where $1 \in G$ is the identity element. We always use normalized cochains below.

From the discussion in the sub-case 'when $\omega$ is in the image of $d_2$' in the last subsection, it suffices to construct a finite Abelian group $A$ and cocycles $b \in Z^{D-1}(G, \hat{A})$, $e \in Z^2(G, A)$ such that $b \cup e = \omega$. As we will see, this can be done almost tautologically.

Consider a free $\mathbb{Z}$-module $\mathsf{A}$ generated by the symbols $\mathsf{x}_{g_1, g_2}$ for $g_{1,2} \in G$, with the understanding that $\mathsf{x}_{1,g} = \mathsf{x}_{g,1} = 0$. One can easily check that the following makes $\mathsf{A}$ a left $G$-module:

$$g_1 \mathsf{x}_{g_2, g_3} = \mathsf{x}_{g_1 g_2, g_3} - \mathsf{x}_{g_1, g_2 g_3} + \mathsf{x}_{g_1, g_2}. \tag{2.37}$$

Then there is a universal 2-cocycle $\mathsf{e} \in Z^2(G, \mathsf{A})$ given by

$$\mathsf{e}(g_1, g_2) = \mathsf{x}_{g_1, g_2}. \tag{2.38}$$

Then we can define a cochain $\mathsf{b} \in C^{D-1}(G, \hat{\mathsf{A}})$ by

$$\langle \mathsf{b}(g_1, g_2, \ldots, g_{D-1}), \mathsf{x}_{g_D, g_{D+1}} \rangle := \omega(g_1, \ldots, g_{D-1}, g_D, g_{D+1}), \tag{2.39}$$

where $\langle \cdot, \cdot \rangle$ is the pairing between $\hat{\mathsf{A}}$ and $\mathsf{A}$. By construction, we have

$$\omega = \mathsf{b} \cup \mathsf{e} \tag{2.40}$$

and also almost by construction, $\mathsf{b}$ is in fact a cocycle $\in Z^{D-1}(G, \hat{\mathsf{A}})$ since

$$\langle (\partial \mathsf{b})(g_1, \ldots, g_{D-1}, h), \mathsf{x}_{g_{D+1}, g_{D+2}} \rangle = (\partial \omega)(g_1, g_2, \ldots, g_{D-1}, h, g_D, g_{D+1}) = 0. \tag{2.41}$$

---

[8]The author thanks Juven C. Wang for explaining the content of Sec. 5 of [13] as a physics proof.

This almost does the job, but $\mathsf{A} \simeq \mathbb{Z}^{(|G|-1)^2}$ is an infinite Abelian group. This small issue can be easily remedied: we take $\Omega \subset U(1)$ be the finite subgroup generated by the value of the cocycle $\omega(g_1, \ldots, g_{D+1})$. Then we take $A$ to be the finite group

$$A = \mathsf{A} \otimes \hat{\Omega} \simeq \hat{\Omega}^{(|G|-1)^2}. \tag{2.42}$$

The tautological cocycle $e \in Z^2(G, A)$ and the cocycle $b \in Z^{D-1}(G, \hat{A})$ can be defined by the same formulas (2.38) and (2.39), and we achieved $\omega = e \cup b$.

# 3 Incorporating 't Hooft defects

## 3.1 An example

Take a 3d theory $T$ with one-form symmetry $A_{[1]} \times G_{[1]}$ where $A = \hat{G}$ is an Abelian group. Suppose the anomaly is given by

$$\int_{Y_4} \boldsymbol{a} \cup \boldsymbol{g}. \tag{3.43}$$

For concreteness, let $A = \mathbb{Z}_n$. Then, in terms of topological defect operators, this just means that the loop operator of charge $k$ for $A$ and the nontrivial loop operator of charge $l$ for $G$ has a nontrivial braiding phase $\exp(2\pi i k l / n)$.

Of course, one example of such theory $T$ is a pure $G = \mathbb{Z}_n$ gauge theory with 't Hooft loops; the line operators associated to $A_{[1]}$ are Wilson lines of $G$, and the line operators associated to $G_{[1]}$ are 't Hooft lines of $G$.

We can consider this situation as a generalization of our discussions so far: we have a trivial extension

$$0 \to A_{[1]} \to A_{[1]} \times G_{[1]} \to G_{[1]} \to 0 \tag{3.44}$$

specified by $\tilde{e} = 0$, with an anomaly specified by $e = \mathrm{id} : H^2(X, G) \to H^2(X, A)$.

Now we can gauge $A_{[1]}$. Our previous discussions applied to this case imply that the resulting gauged theory $T/A$ has a symmetry

$$0 \to \hat{A}_{[0]} \to \underline{\Gamma} \to G_{[1]} \to 0, \tag{3.45}$$

whose extension is specified by the same $e$, with the zero anomaly $\tilde{e} = 0$. What does this abstract nonsense mean, in concrete terms?

We first note that we took $\hat{A} = G$. Then, a background field for $\underline{\Gamma}$ is a pair $(\boldsymbol{a}, \boldsymbol{g})$ where $\boldsymbol{g} \in H^2(X, G) = H_1(X, G)$ while $\boldsymbol{a} \in C^1(X, \hat{A})$ is such that

$$\partial \boldsymbol{a} = e(\boldsymbol{g}) = \boldsymbol{g}. \tag{3.46}$$

This just means that $\boldsymbol{a}$ is a surface in $X$ whose boundary is $\boldsymbol{g}$.

When $T$ is the pure $\mathbb{Z}_n$ gauge theory with 't Hooft loops, $T/\hat{A}$ is a trivial theory with a $\mathbb{Z}_n$ symmetry with 't Hooft loops. $\boldsymbol{a}$ is the domain wall representing the $\mathbb{Z}_n$ action; $\boldsymbol{g}$ specifies the 't Hooft loops around which we have nontrivial $\mathbb{Z}_n$ holonomy; and the relation (3.46) means that the 't Hooft loops are boundaries of the domain walls.

Summarizing, the relation between 't Hooft loops of a flavor symmetry and domain walls representing the flavor symmetry can be thought of as an extension of a 1-form symmetry by a 0-form symmetry. It is not clear what this reinterpretation buys us at present, but in Sec. 3.3 we at least point out a somewhat curious phenomenon.

## 3.2 Extension of $G_{[n]}$ by $A_{[m]}$

Before getting there, let us state the general situation of an extension of $n$-form $G$ symmetry by an $m$-form $A$ symmetry. Here $G$ is also assumed to be Abelian when $n > 0$. We use the basics of Eilenberg-Mac Lane spaces below, which is briefly reviewed in Appendix A.1.

We pick two elements

$$e \in H^{m+2}(K(G, n+1), A), \qquad \tilde{e} \in H^{D-m}(K(G, n+1), \hat{A}) \tag{3.47}$$

satisfying

$$e \cup \tilde{e} = 0 \in H^{D+2}(K(G, n+1), U(1)). \tag{3.48}$$

We choose a cochain

$$\omega \in C^{D+1}(K(G, n+1), U(1)) \tag{3.49}$$

such that

$$\partial \omega = e \cup \tilde{e}. \tag{3.50}$$

To this, we associate an extension

$$0 \to A_{[m]} \to \underline{\Gamma} \to G_{[n]} \to 0, \tag{3.51}$$

whose extension class is specified by $e$. Concretely, this means that a background field for $\underline{\Gamma}$ is a pair $(\boldsymbol{a}, \boldsymbol{g})$ where $\boldsymbol{a} \in C^{m+1}(X, A)$, $\boldsymbol{g} \in H^{n+1}(X, G)$ such that

$$\partial \boldsymbol{a} = e(\boldsymbol{g}), \tag{3.52}$$

where $e$ is now thought of as a cohomology operation

$$e : H^{n+1}(-, G) \to H^{m+2}(-, A). \tag{3.53}$$

Similarly, we have another extension

$$0 \to \hat{A}_{[D-m-2]} \to \underline{\tilde{\Gamma}} \to G_{[n]} \to 0 \tag{3.54}$$

for which we think of $\tilde{e}$ as a cohomology operation

$$\tilde{e} : H^{n+1}(-, G) \to H^{D-m}(-, \hat{A}) \tag{3.55}$$

so that the background fields satisfy

$$\partial \hat{\boldsymbol{a}} = \tilde{e}(\boldsymbol{g}). \tag{3.56}$$

We define an anomaly for $\underline{\Gamma}$ by the formula

$$\alpha(\tilde{e}, \omega) := \int_{Y_{D+1}} (\boldsymbol{a} \cup \tilde{e}(\boldsymbol{g}) - \omega(\boldsymbol{g})) \tag{3.57}$$

and an anomaly for $\underline{\tilde{\Gamma}}$ by the formula

$$\tilde{\alpha}(e, \omega) := \int_{Y_{D+1}} (\hat{\boldsymbol{a}} \cup e(\boldsymbol{g}) - \omega(\boldsymbol{g})). \tag{3.58}$$

We then have:

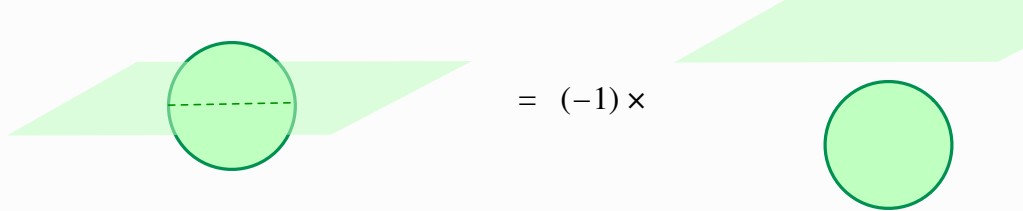

Figure 7: A curious skein relation in a 3d theory. Here a thik line represents a $\mathbb{Z}_2$ line which is a boundary of a $\mathbb{Z}_2$ wall. When it protrudes across another $\mathbb{Z}_2$ wall, there is a minus sign.

*Given $e$, $\tilde{e}$ and $\omega$ as above, a theory with the symmetry*

$$0 \to A_{[m]} \to \underline{\Gamma} \to G_{[n]} \to 0, \tag{3.59}$$

*whose extension is $e$ with an anomaly $\alpha(\tilde{e}, \omega)$ and a theory with the symmetry*

$$0 \to \hat{A}_{[D-m-2]} \to \underline{\tilde{\Gamma}} \to G_{[n]} \to 0, \tag{3.60}$$

*whose extension is $\tilde{e}$ with an anomaly $\tilde{\alpha}(e, \omega)$ are exchanged by the gauging of $A$ and $\hat{A}$.*

The derivation is again entirely analogous to the previous ones given. In Appendix A.3, we discuss the classifying space for $\underline{\Gamma}$ and gives an interpretation of the condition (3.48) in that language.

## 3.3 A curious phenomenon

In Sec. 3.1 we considered the defects in a 3d $\mathbb{Z}_n$ gauge theory, which has loop operators labeled by $\mathbb{Z}_n \times \mathbb{Z}_n$. We saw that when $\mathbb{Z}_n$ is ungauged, we found that a $\mathbb{Z}_n$ domain wall can end on a $\mathbb{Z}_n$ 't Hooft loop.

Here we consider a 3d theory with a $\mathbb{Z}_4$ 1-form symmetry instead, with the anomaly given by the requirement that the topological line operator labeled by the generator of $\mathbb{Z}_4$ has a braiding phase $-1$ with itself. An example is to take two copies of $U(1)_4$ Chern-Simons theory and identify the two $\mathbb{Z}_4$ 1-form symmetries. Now, $\mathbb{Z}_4$ fits in an extension

$$0 \to (\mathbb{Z}_2)_{[1]} \to (\mathbb{Z}_4)_{[1]} \to (\mathbb{Z}_2)_{[1]} \to 0, \tag{3.61}$$

and we find that $\mathbb{Z}_2 \subset \mathbb{Z}_4$ is non-anomalous and can be gauged. This fits in the framework of the last subsection.

The gauged theory then has a mixed symmetry

$$0 \to (\mathbb{Z}_2)_{[0]} \to \underline{\Gamma} \to (\mathbb{Z}_2)_{[1]} \to 0, \tag{3.62}$$

with a nontrivial extension and a nontrivial anomaly. The extension is the same as we discussed in (3.45) in Sec. 3.1, and it says that the $\mathbb{Z}_2$ domain wall can end on the $\mathbb{Z}_2$ line operator. The anomaly appears as the fact that the intersection points of $\mathbb{Z}_2$ line operators on the $\mathbb{Z}_2$ domain wall has a nontrivial "braiding" relation $-1$, see Fig. 7.

Again the author does not know how to make of this phase, but it might be useful to remember that there can be such a phase, between the symmetry-implementing domain walls and the 't Hooft loops.

# 4 Subtler effects

Let us come back to the case when the symmetry $\Gamma$ of a theory $T$ is an ordinary group fitting in an extension

$$0 \to A \to \Gamma \to G \to 0. \tag{4.63}$$

So far we considered what happens if we gauge $A$ when the anomaly of $\Gamma$ has a very specific form (2.22), essentially determined by a class $\tilde{e} \in H^D(G, \hat{A})$.

In general, an arbitrary anomaly of the group $\Gamma$, or equivalently an element of $H^{D+1}(\Gamma, U(1))$, can be studied in terms of $H^p(G, H^q(A, U(1)))$ with $p + q = D + 1$. The systematics is provided by the LHS spectral sequence, briefly reviewed in Sec. A.2. So far, we are only utilizing two components

$$H^{D+1}(G, H^0(A, U(1))) = H^{D+1}(G, U(1)) \tag{4.64}$$

and

$$H^D(G, H^1(A, U(1))) = H^D(G, \hat{A}). \tag{4.65}$$

We saw the latter includes the infamous $H^3(G, \hat{A})$ obstruction which appears in the discussion of how a $G$ symmetry would act on anyons in 3d TQFTs.

These are however only the first two steps in the long ladder. Other $H^p(G, H^q(A, U(1)))$ with $q \geq 2$ would provide more esoteric behaviors among topological defect operators in the gauged theory $T/A$. When $D = 2$, everything is understood and can be phrased in the language of fusion categories, as reviewed e.g. [22]. There, the only remaining component is $H^1(G, H^2(A, U(1)))$, and it is known that a nonzero element in it would make the symmetry of the gauged theory $T/A$ to be non-group-like.

In this last section,[9] we restrict our attention to the direct product $\Gamma = A \times G$, and consider the effect of an anomaly determined by an element $t \in H^{D-1}(G, H^2(A, U(1)))$. Namely, we consider the case where the total anomaly of the original theory $T$ is given by

$$\int_{Y_{D+1}} \boldsymbol{a} \cup \boldsymbol{a} \cup t(\boldsymbol{g}), \tag{4.66}$$

where we first consider the integrand as the $(D+1)$-cocycle valued in $A \otimes_{\mathbb{Z}} A \otimes_{\mathbb{Z}} H^2(A, U(1))$, then use the fact that elements of $H^2(A, U(1))$ can naturally be thought of as elements of $\hat{A} \otimes_{\mathbb{Z}} \hat{A}$ which is skew-symmetric to regard it as a U(1)-valued $(D+1)$-cocycle.

Equivalently, the element $t$ characterize the anomaly in the following manner. Put the theory on $M_{D-1}$ with a $G$-bundle $\boldsymbol{g}$ on it, reduce it to a one-dimensional theory, and consider it as a theory with $A$ symmetry. Then its $A$ anomaly is given by the element

$$\int_{M_{D-1}} t(\boldsymbol{g}) \in H^2(A, U(1)). \tag{4.67}$$

## 4.1 $D = 2$

Let us first consider the case $D = 2$. We are going to gauge the symmetry $A$. When $t$ is nonzero, there is an $S^1$ compactification with nontrivial $\boldsymbol{g}$ to one dimension such that the gauge anomaly (4.67) is nonzero. In other words, there is a mixed flavor-gauge anomaly between the gauge group $A$ and the flavor group $G$. This usually means that $G$ is not a symmetry anymore. If we accept defeat in this manner, the symmetry of the gauged theory $T/A$ is just $\hat{A}$, and we can no longer recover the whole symmetry $A \times G$ by re-gauging $\hat{A}$ of the theory $T/A$. We would not accept this.

---

[9]The author thanks Kantaro Ohmori for convincing me to write up the content of this section.

Still, mathematicians have found a way to consider more topological defect operators in the gauged theory $T/A$ so that re-gauging $\hat{A}$ recovers the whole symmetry $A \times G$. They had their own motivation in the construction. Here we give a physics interpretation.

When the spacetime is one dimensional, it is very easy to give a topological theory with symmetry $A$ with a given anomaly $\tau \in H^2(A, U(1))$: it is just a fancy name for a projective representation of $A$ whose projective phase is characterized by $\tau$. Therefore, we just add a projective representation of $A$ on top of a domain wall implementing $g \in G$ such that $t(g) \in H^2(A, U(1))$ is nonzero, to cancel the anomaly on the domain wall. This is a natural generalization of the fact that on a domain wall for $\mathrm{id} \in G$, we consider a genuine representation of $A$, i.e. they are ordinary Wilson lines labeled by elements of $\hat{A}$.

Summarizing, the topological line operators of the gauged theory $T/A$ are labeled by the pairs of the form

$$(\rho, g) \in (\mathrm{Rep}_{t(g)} A, G), \tag{4.68}$$

where $\mathrm{Rep}_{t(g)} A$ denotes the set of projective representations of $A$ whose phase is characterized by $t(g) \in H^2(A, U(1))$. These line operators fuse according to the following rule:

$$(\rho, g) \otimes (\sigma, h) = (\rho \otimes \sigma, gh). \tag{4.69}$$

This rule is consistent, since $\rho \otimes \sigma$ has the projective phase characterized by $t(gh) = t(g)t(h)$.

As an example, consider a theory $T$ with the symmetry $A \times G = (\mathbb{Z}_2 \times \mathbb{Z}_2) \times \mathbb{Z}_2$ with the anomaly

$$\int_{Y_3} \boldsymbol{a}_1 \cup \boldsymbol{a}_2 \cup \boldsymbol{g}. \tag{4.70}$$

We gauge $A = \mathbb{Z}_2 \times \mathbb{Z}_2$. Denote the elements of $G = \mathbb{Z}_2$ by 0 and 1. Then the labels for the irreducible lines are either

- $(\hat{a}, 0)$ where $\hat{a} \in \hat{A} = \hat{\mathbb{Z}}_2 \times \hat{\mathbb{Z}}_2$, or

- $(\rho, 1)$ where $\rho$ is the unique irreducible non-trivial projective representation of $\mathbb{Z}_2 \times \mathbb{Z}_2$, which is two-dimensional.

Note that this set of labels and the resulting fusion rule are exactly the same as that of the irreducible representations of the dihedral group $D_8$ with eight elements. This is as it should be, since $D_8$ is an extension

$$0 \to \hat{G} = \hat{\mathbb{Z}}_2 \to D_8 \to A = \mathbb{Z}_2 \times \mathbb{Z}_2 \to 0, \tag{4.71}$$

and therefore an irreducible representation of $D_8$ can be thought of as possibly projective representations of $\mathbb{Z}_2 \times \mathbb{Z}_2$, whose projective phase is determined by how the first $\hat{\mathbb{Z}}_2$ is represented.

This observation has a natural explanation. Consider the same theory $T$ but gauge $G$ instead. Here we can apply the result in Sec. 2.2; we see that $T' = T/G$ has the symmetry $D_8$, which contains $\hat{G}$ as a subgroup. Then the theory $T/A$ we are considering is in fact

$$T/A = (T'/\hat{G})/A = T'/D_8 \tag{4.72}$$

since the $D_8$ symmetry of $T'$ is composed of $\hat{G}$ and $A$. Therefore the theory $T/A$ should have all the Wilson lines of $D_8$.[10]

---

[10]This decomposition of an orbifold by a solvable non-Abelian group into repeated applications of Abelian orbifolds was utilized before in the context of the study of D-branes at orbifold singularities [36].

## 4.2   $D > 2$

Now the extension to $D > 2$ is clear. We start from the theory $T$ with the symmetry $A \times G$, with the anomaly specified by $t \in H^{D-1}(G, H^2(A, \mathrm{U}(1)))$. Given a $G$-bundle $\boldsymbol{g}$, we have one-dimensional subloci specified by $t(\boldsymbol{g})$. Such a locus can be visualized as the place where $(D-1)$ domain walls $g_1, \ldots, g_{D-1} \in G$ meet to form a new wall labeled by $g' = g_1 \cdots g_{D-1}$, which determine an element

$$t(g_1, \ldots, g_{D-1}) \in H^2(A, \mathrm{U}(1)). \tag{4.73}$$

We simply demand that such a one-dimensional locus should carry a projective representation of $A$ whose projective phase is characterized by the element (4.73).

When $D = 3$, this construction should give rise to a 2-category as described in [4]. Since the nontrivial projective representations of an Abelian groups are not one dimensional and therefore not invertible, this gives an explicit example of such a 2-category which is not a 2-group.

## 4.3   Toward even subtler effects

So far we discussed the effect of elements of $t \in H^{D-1}(G, H^2(A, \mathrm{U}(1)))$ in the gauged theory $T/A$. We saw that each one-dimensional locus at the intersection of domain walls implementing the $G$ action can naively carry a nonzero gauge anomaly determined by $t$, which is then canceled by putting a projective representation of $A$.

The simpler effect from elements of $\tilde{e} \in H^D(G, H^1(A, \mathrm{U}(1)))$, which we examined in more detail in the earlier part of the note, can be phrased in a similar manner: each zero-dimensional locus at the intersection of domain walls of $G$ can naively carry a "nonzero gauge anomaly" of a zero-dimensional $A$ gauge theory, which is just a nonzero excess charge under $A$ of the point being considered. This is canceled by attaching a Wilson line of $A$.

From these considerations, we can describe, at least schematically, the effect of elements of $t \in H^{D-d}(G, H^{d+1}(A, \mathrm{U}(1)))$ would have on the structure of the topological defect operators. Namely, on a dimension-$d$ locus where $D - d$ domain walls implementing $g_1, \ldots, g_{D-d}$ meet to form a domain wall labeled by $g' = g_1 \cdots g_{D-d}$, there is naively a gauge anomaly given by

$$t(g_1, \ldots, g_{D-d}) \in H^{d+1}(A, \mathrm{U}(1)). \tag{4.74}$$

We cancel this by introducing there some TQFT with symmetry $A$ whose anomaly is given by the expression (4.74) above. As we recalled in Sec. 2.6, it was argued that there is always such a TQFT, so we can keep arbitrary walls implementing the $G$ action in the theory, by putting these anomalous lower-dimensional TQFTs on the walls.

What complicates the cases $d \geq 2$ as compared to the cases $d = 0, 1$ is the following. For $d = 0, 1$, the full list of $d$-dimensional TQFTs with $A$ symmetry with a given anomaly in $H^{d+1}(A, \mathrm{U}(1))$ is known, and therefore the structure of the topological defect operators of the gauged theory can be worked out concretely. For $d \geq 2$, our understanding of the full list of such TQFTs with $A$ symmetry with a given anomaly is still rudimentary, thus precluding us from a full description of the topological description of the gauged theory.

# Acknowledgments

The author thanks Nati Seiberg for suggesting him to study the so-called $H^3(G, \hat{A})$ "anomaly" in the first place, during the Strings 2017 conference. It is also a pleasure for the author to thank Francesco Benini, Clay Córdova, Kantaro Ohmori, Nati Seiberg, Po-Shen Hsin and Juven C. Wang for inspiring discussions. The author also thanks Kantaro Ohmori for suggesting the

notation $G_{[n]}$ for a higher-form symmetry, Edward Witten for suggesting an improvement in the presentation in an earlier version of the paper, and an anonymous referee for detailed comments which improved the paper greatly. The author is partially supported in part byJSPS KAKENHI Grant-in-Aid (Wakate-A), No.17H04837 and JSPS KAKENHI Grant-in-Aid (Kiban-S), No.16H06335, and also supported in part by WPI Initiative, MEXT, Japan at IPMU, the University of Tokyo.

# A    Some algebraic topology

In this subsection we very briefly review the basics of algebraic topology needed in the note. For more details, please consult standard textbooks, e.g. [37].

## A.1    Eilenberg-Mac Lane spaces

The Eilenberg-Mac Lane space $K(G, n+1)$ is a space (defined up to homotopy) such that the only nontrivial homotopy group is $\pi_{n+1} = G$. When $n > 0$ this requires that $G$ is Abelian.

The space $K(G, n+1)$ serves as the classifying space for the background fields for the $n$-form finite symmetry group $G$,[11] which means the following:

When $n = 0$, the background field for a finite group symmetry $G$ on a space $X$ is a $G$-bundle. Distinct $G$-bundles on $X$ are in bijective correspondence with the homotopy class of maps $[X, K(G, 1)]$. $K(G, 1)$ is also denoted as $BG$, the classifying space of $G$. When $n \geq 0$ and $G$ Abelian, the background field for an $n$-form $G$ symmetry on a space $X$ is an element of cohomology group $H^{n+1}(X, G)$. Again, distinct elements are in bijective correspondence with the homotopy class of maps $[X, K(G, n+1)]$. Similarly, we can denote $K(G, n+1)$ as $B^n G$.

This means that cohomology classes on $K(G, n+1)$ are characteristic classes for a background field for an $n$-form symmetry $G$. Indeed, given a background field on $X$, regard it as a (homotopy class of a) map $\boldsymbol{g} \in [X, K(G, n+1)]$. Pick an $M$-valued cohomology class $\alpha \in H^d(K(G, n+1), M)$. Then we can pull back $M$ via $\boldsymbol{g}$ to $X$ and find a cohomology class $\boldsymbol{g}^*(\alpha) \in H^d(X, M)$. We denote it as $\alpha(\boldsymbol{g})$, regarding $\alpha$ as an operation

$$\alpha : [-, K(G, n+1)] \to H^d(-, M). \tag{A.75}$$

When $G$ is Abelian, this means that $\alpha$ is a cohomology operation

$$\alpha : H^{n+1}(-, G) \to H^d(-, M). \tag{A.76}$$

The cohomology of the Eilenberg-Mac Lane space $H^d(K(G, 1), M)$ is also known as the group cohomology and is also written as $H^d(G, M)$. It has a purely algebraic definition which we freely used in this note.

## A.2    Lyndon-Hochschild-Serre spectral sequence

Given an Abelian extension of groups $0 \to A \to \Gamma \to G \to 0$, the Lyndon-Hochschild-Serre (LHS) spectral sequence is a spectral sequence converging to $H^{p+q}(\Gamma, M)$ whose second page is given by $E_2^{p,q} = H^p(G, H^q(A, M))$. Using the classifying space, this is just the Leray-Serre spectral sequence associated to the fibration

$$BA \to B\Gamma \to BG \tag{A.77}$$

---

[11]This shift by 1 is somewhat regrettable but cannot be avoided now that there is already a standard terminology on the physics side. This is somewhat similar to the situation of a D$p$ brane having a $(p+1)$-dimensional worldvolume.

but there is also a purely algebraic description which is sometimes handier for concrete computations.

Both the topological and algebraic descriptions can be found in the original paper by Hochschild and Serre [38],[12] which contains the result that when the action of $\Gamma$ on $M$ is trivial, the differential

$$d_2 : E_2^{p,1} \to E_2^{p+2,0} \tag{A.78}$$

or equivalently

$$d_2 : H^p(G, \text{Hom}(A, M)) \to H^{p+2}(G, M) \tag{A.79}$$

is given by the cup product by the extension class $e \in H^2(G, A)$.

### A.3  Classifying space for the mixed symmetry

For an $n$-form symmetry $G_{[n]}$, the classifying space is $B(G_{[n]}) := B^{n+1}G := K(G, n+1)$, the Eilenberg-Mac Lane space. Given an extension

$$0 \to A_{[m]} \to \underline{\Gamma} \to G_{[n]} \to 0 \tag{A.80}$$

the classifying space is a fibration

$$K(A, m+1) \to B\underline{\Gamma} \to K(G, n+1). \tag{A.81}$$

Taking the homotopy cofiber, we have another fibration

$$B\underline{\Gamma} \to K(G, n+1) \to K(A, m+2) \tag{A.82}$$

by which we have a class $e \in H^{m+2}(K(G, n+1), A)$ called the Postnikov class or the Postnikov $k$-invariant. This class $e$ in fact classifies the fibration (A.81).

The Leray-Serre spectral sequence for the fibration (A.81) is a spectral sequence whose second page is $E_2^{p,q} = H^p(K(G, n+1), H^q(K(A, m+1), M))$ converging to $H^{p+q}(B\underline{\Gamma}, U(1))$. A routine modification of the argument in [38] which uses the multiplicative property of the spectral sequence says that

$$d_{m+2} : E_{m+2}^{p,m+1} \to E_{m+2}^{p+m+2,0} \tag{A.83}$$

is given by the cup product by the extension class $e$.

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
