# Peer review of "On gauging finite subgroups"

_SciPost Physics, doi:SciPost Phys. 8, 015 (2020)_

## Round 2 · Referee Report · Anonymous (Referee 1) · 2018-7-11

Strengths

1. clearly written
2. gives nice overview of important subject of generalized symmetry
3. finds good balance between general theory and concrete examples

Weaknesses

none

Report

The paper deals with the subject of generalized symmetries in the context of (topological) quantum field theories with defects. More concretely, it studies the gauging of finite subgroups of symmetry subgroups, bringing together ideas and results from the perspective of domain walls and algebraic topology, and a clear discussion of how anomalies relate to this description. I strongly recommend publication of the paper, and I suggest to address the points raised below as the author sees fit.

Requested changes

1. Footnote 1: Why does the doubled coset appear as a label set for domain walls?
2. Footnote 1: Does "consistent set of topological defects" mean that it is closed with respect to fusion?
3. Page 3, Notations and conventions: Please give a brief reminder on "n-form symmetry". How does it relate to n-groupoids? (This is connected to item 14 below.)
4. Page 3, Notations and conventions: At first I was confused by the dimensions of the spaces X and Y. Please briefly explain why the dimension D+1 appears even though spacetime is D-dimensional.
5. Page 4, last paragraph: Its seems that this discussion is also a partial TFT interpretation of Pachner's theorem on D-dimensional triangulations. Has the author considered this in more detail?
6. Page 6, boxed result: What exactly is Y here, and how does the result depend on it? Why was it not mentioned more prominently earlier in Section 2?
7. Page 7: Why does gh in G serve as a source (and not as target) of the domain wall?
8. Page 8: Please give more details for the last paragraph before Section 2.3.
9. Footnote 5: I would have thought that the non-commutative algebra "defines" the space M indirectly, not directly.
10. Page 9, last item: One could argue that the extension to a 3-functor from BG to the 3-category of certain fusion categories and their bimodule categories (as studied by Etingof, Nikshych and Ostrik) is even more relevant in TQFT; in this case there is also an H^4-obstruction.
11. Page 14, second paragraph of Section 2.7: Please give a reference for the well-known result.
12. Page 17, first paragraph of Section 3.3: In my copy of the paper Section 3.1 treats the case Z_n, not only Z_2.
13. Page 18, last paragraph: What do the \otimes-symbols mean?
14. Page 23: Is G_{[n]} really equal to B^nG?
15. Typos:
- "Gaitto" on Page 2
- "two analysis" on Page 3
- "3)-chain" on Page 4
- "as also as" on Page 7
- "we know have" on Page 8
- "described blow" on Page 11

---

## Round 3 · Author Response

I am very sorry that my extremely slow reply to the very detailed referee report.
I summarized below the changes made according to the suggestions by the referee.

---

## Round 3 · List of Changes

1,2: A reference was added.

3: I added the information that an $n$-form symmetry comes with a map to $K(G,n+1)$.

4: This is due to the fact that the general yoga says that the anomaly of a $D$-dimensional theory is captured by a $(D+1)$-dimensional invertible topological phase. I added a few references on this point. A short paragraph was added just before "organization of the note" to mention this important fact.

5: This was studied in the past; two references were added. I also explicitly mentioned which of the spacetimes $X_D$ and $Y_{D+1}$ were considered in various steps involved.

6: Indeed I should have mentioned the role of $Y$ more prominently. This I think is covered by the added paragraph mentioned in the point 4 above, and the clarification mentioned in the point 5 above.

7: The word "source" was used in the sense of the "source of (electric and other) charges" used in physics, and not in the sense of the source and the target. The word is changed to the "boundary", which more accurately captures the situation and should cause less confusion.

8: I tried to clarify the explanation by adding a few sentences describing an additional intermediate step, so that the original Figures 3,4,5 were replaced by the new Figures 3,4,5,6.

9: I agree with the referee's comment. I tried to clarify the footnote.

10: I agree with the referee's comment. A paragraph and two references were added.

11: The original article by Eilenberg and Mac Lane was added.

12: It was a carry-over from an earlier version of the note; I thank the referee for the careful reading. They are corrected.

13: They are tensor product as $\mathbb{Z}$-modules. This fact was made explicit.

14: Essentially, yes, but I would like to keep the notation $G_{[n]}$ as a shorthand for the (ordinary) group $G$ regarded as an $n$-form symmetry.

15: I'd like to thank again the referee for the careful reading. They are all corrected.

You are currently on this page

Resubmission 1712.09542v3 on 24 December 2019

---

## Editorial Decision

published